# Evaluation and Application of an Engineering Calculation Method of the Static Performance of an Aerostatic Journal Bearing with Multiple Orifice-Type Restrictors

Yangong Wu, Zheng Qiao, Wentao Chen, Jiadai Xue and Bo Wang *

Center for Precision Engineering, Harbin Institute of Technology, Harbin 150001, China
* Correspondence: bradywang@hit.edu.cn

**Abstract:** A simplified calculation method is evaluated to calculate the static performance of an aerostatic journal bearing with multiple orifice-type restrictors. This method adopts a one-dimension flow assumption and is a fast calculation method to design journal bearings in engineering by directly linking the structural parameters and performance parameters affecting radial bearings with nonlinear equations. In addition, this method is verified with computational fluid dynamics by two actual case studies, and it is found that the LCC difference between those two methods is less than 5% for a 200 mm diameter spindle, and less than 10% for a 100 mm diameter spindle. Subsequently, the influence of a key parameter $\zeta_i$ on the static performance of journal bearings is explained theoretically. This method is much easier and more intuitive compared with numerical computational methods. Furthermore, it promotes the application of aerostatic journal bearings.

**Keywords:** simplified calculation method; performance analysis; aerostatic journal bearing





## 1. Introduction

Owing to the absence of mechanical contact between moving parts, it is possible to obtain extremely high accuracy and high speed from a motion system when supported by aerostatic bearing. Therefore, aerostatic bearings have been commonly utilized in positioning systems, and precision machining equipment, etc. [1–3]. Most of the film flow field can be directly described by the Reynolds equation; therefore, the core work of gas bearing performance calculation is to solve the Reynolds equation according to the given input parameters (geometry, lubricant properties, working conditions, boundary conditions, etc.). The basic problem of fluid-film lubricated bearings is to determine the pressure distribution under various working conditions. On the basis of obtaining the pressure distribution, the performance parameters of the bearing, such as stiffness and load carrying capacity (LCC), which are of great importance during and after the design, can be obtained by integrating/differentiating the pressure field on the air film surface.

Lots of calculation methods have been suggested to address the nonlinear Reynolds equation governing pressure distribution in aerostatic bearing and static performance [4–6]. In addition to these computational methods for solving the general Reynolds equation, there have been many developments in computational methods for aerostatic bearings with external restrictors. There are two methods for solving the characteristics of aerostatic bearings: analytical and numerical. The analytical solution can roughly obtain the bearing characteristics, point out important variables and parameters, narrow the scope of the attention of variables, and help to intuitively understand the physical situation. Powell proposed the gauge-pressure ratio method, which gives the formula for the static performance of aerostatic bearings [7]. The gauge-pressure ratio method borrows the symbol of hydrostatic bearings, and the compressibility of gas is not properly handled, so it is necessary to manually consult the table to obtain the bearing characteristics. Li et al. described a simplified calculation method of aerostatic thrust bearing with multiple pocketed orifice-type

restrictors (MPOTRs); however, this simplified model has not been discussed for aerostatic journal bearing [8]. Tang and Gross summarized the one-dimensional flow model for thrust and journal bearings in the early stage of the development of lubrication mechanics, but it is limited by computer performance and lacks further analysis of key processes [9]. Jason described the same flow model for journal bearings with one-row orifices [10]. Belforte et al. presented two lumped-parameter models to rapidly obtain the static characteristic of gas bearings [11]. It is worthwhile to mention that since the steady-state Reynolds equation in a constant-thickness gas film region is the Laplace equation, Mori et al. used the complex potential method to obtain an analytical formula for radial bearing, the porous planar thrust bearing and the air-floating pad without eccentricity [12,13].

The above engineering calculation method is still widely used, but it can only solve the approximate calculation problem of the regular geometric boundary and the thickness of the gas film. To obtain more accurate prediction results, numerical methods such as the finite element method (FEM) and finite difference method (FDM) have developed with the rise in computer technology. Using the variational principle of operator equations, Reddi deducted the direct and incremental variational formulations of two-dimensional incompressible and compressible steady-state Reynolds equations, and applied FEM to the thin-film lubrication problem for the first time [14,15]. After that, Wadhwa and Nguyen et al. further developed the relevant theory of FEM to solve the performance of gas dynamic pressure bearing [16,17]; Wang and co-workers presented a computational method for solving the isothermal compressible Reynolds equation to estimate mass flow rate (MFR), friction and LCC. They linearized the Reynolds equation with the Newton method and solved it using an iterative successive relaxation method [18]. Lo and colleagues proposed a finite element method to evaluate the performance of a high-speed gas journal bearing. The dimensionless Reynolds equation was derived from the simplified dimensionless Navier-Stokes (NS) equations, discretized using the Newton method and solved by means of an iterative cutting procedure [19]. Wadhwa and Cui calculated the static characteristics of common double-row orifice radial bearings [20,21]. Li et al. calculated the static characteristics of the orifice-type aerostatic guideway [22]. Charki and Cui solved the static characteristics of the orifice-type aerostatic ball bearings [23,24]. Wu et al. found that inappropriate boundary settings can cause significant numerical errors if the grid-dependent condition is ignored, and their calculation results reveal that those numerical models with a single-source node are grid-dependent, and auxiliary sources have proved to be an effective technique to reduce single-source numerical error [25]. Chen et al. analyzed the impact on the bearing of orifice blockage in actual working conditions; the air film pressure distribution, load capacity, and tilting moment were obtained and simulation results indicate that the air film pressure decreases significantly in the blocked area of adjacent orifices [26]. In addition, Rajdeep et al. seems to be the first to solve the Reynolds equation adopting the finite volume method [27].

In addition to only calculating the air film area, computational fluid dynamics (CFD) has also been introduced to figure out bearings' performance precisely because it solves NS equations rather than the Reynolds equation, which takes all the air paths including inlet ports, orifices, and air films into consideration. Cui et al. used Fluent software to calculate the static performance of porous aerostatic thrust bearings [28]. Wu et al. simulated the static performance of radial and thrust orifice-type bearings in a spindle designed for ultra-precision machine tools [29]. Belforte et al. gave a practical flow correction coefficient formula of small holes through CFD simulation and experiments, and unified the flow calculation method of self-contained and orifice restriction [30–33]. Renn, Yoshimoto, Eleshaky and Chang et al. carried out simulations or experiments on the static characteristics of a single-hole restriction thrust bearing [34–37], aiming to understand what happened between the region of orifice and gas film. Ting et al. utilized a CFD software to evaluate the performance of a radial–axial integrated cup-shaped aerostatic bearing for the miniature turbine of a dental handpiece [38], and Li et al. presented and simulated a novel aerostatic bearing with back-flow channels, which are designed to connect the feed pocket and

low-pressure region of the bearing clearance directly [39]. CFD is of high computational accuracy, but it is time-consuming to model, computationally inefficient, and sometimes plagued by stability problems. A recent developing hybrid modeling method was proposed by Neves, Chen and Song, which uses CFD to accurately simulate the performance of orifice, and, at the same time, makes full use of FEM to efficiently calculate the Reynolds equation [40–42].

When it comes to numerical methods, complicated iterations are needed, the calculating procedure is sophisticated, and the calculating process is time-consuming, which makes it complex and burdensome to analyze the effect of a bearings' parameters on its performance on account of the non-intuitiveness of the calculation process [43]. In comparison, with FEM, it is a little easier to optimize a bearing's parameters more conveniently, but FEM requires regenerating meshes and rearranging the program with varying design parameters [44].

In this study, a simplified calculation method is proposed to simplify the calculation of pressure in aerostatic journal bearing with MPOTRs, and the influence of the bearing's parameters on its performance is explored. Additionally, an aerostatic bearing with specific parameters is calculated by CFD to verify the method. Furthermore, the effect of $\zeta_i$ on a bearing's performance section is used to determine the impact of the bearing's parameters on its performance, and some conclusions are given.

## 2. Principle of the Simplified Calculation Method

Figure 1 shows an aerostatic journal bearing with MPOTRs. Both ends of external circumference with diameter $D$ are open. Hence, 2 atmospheric boundaries are found. $L$ is the total length where gas film exists, $l$ is the distance between $N$ orifices' location and gas film edge. To ensure that the restrictor area is orifice sectional area, the air chamber is connected to the orifice. A diffusion effect is observed immediately when the gas effuses out of an orifice and the gas distribution becomes gradually uniform. Diffusion effect can be ignored if the orifices are linked by grooves [45]. In addition, a detailed description of orifice can be found in [46].

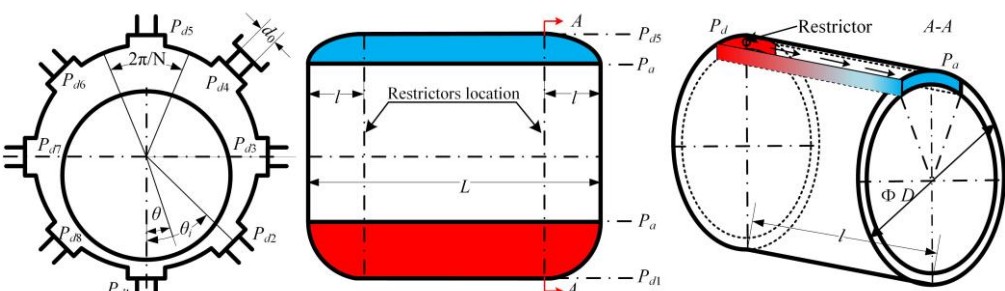

**Figure 1.** Configuration of a journal bearing with MPOTRs.

To simulate a genuine situation, the following assumptions were made in the simplified calculation method. There is no flow in the circumferential direction. In other words, the gas can distribute uniformly in every $2\pi/N$ section edge and the pressure from $P_d$ decreases to $P_a$ smoothly. Lubricant gas appears to be ideal gas, which is compressible and isothermal, because the heat transfer between gas and solid could be neglected. The gas inside bearing is of low Reynolds number in the clearance because the film thickness $h_0$ is within tens of micrometers. Therefore, the cylindrical area can expand into a rectangle, which makes the 3-D cylindrical surface problem into a plane problem. The width $b$ of 1-D gas film in Figure 2 is equal to $\pi D/N$ in Figure 1.

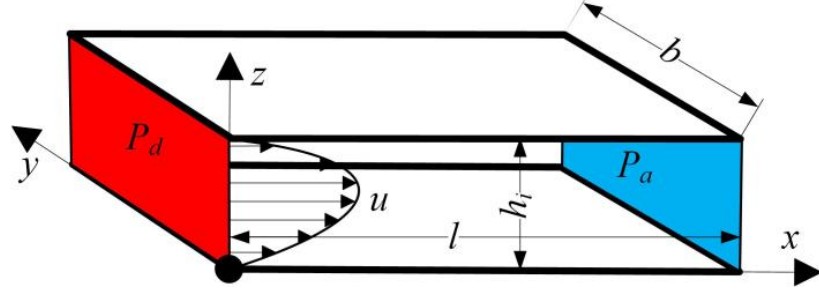

**Figure 2.** Flow model in each part of $i$th section.

According to the assumptions above, the bearing pressure $P$ can satisfy Equations (1)–(3) in the Cartesian coordinates displayed in Figures 1 and 2.

$$\frac{\partial P}{\partial x} = \eta \frac{\partial^2 u}{\partial z^2} \tag{1}$$

$$\frac{\partial P}{\partial y} = 0 \tag{2}$$

$$\frac{\partial P}{\partial z} = 0 \tag{3}$$

Equations (2) and (3) demonstrate that $P$ is not connected to $y$ and $z$. Thus, Equation (1) is expressed as

$$\frac{\mathrm{d}P}{\mathrm{d}x} = \eta \frac{\mathrm{d}^2 u}{\mathrm{d}z^2} \tag{4}$$

Equation (4) is the simplified Reynolds equation under 1-D hypnosis. Supposing the film thickness in each section of the bearing surface is the same, MFR of $i$th section is

$$\dot{m}_i = A_o \varnothing P_0 \sqrt{\frac{2\rho_a}{P_a}} \varphi(\beta_i) \tag{5}$$

where $A_0$ is the section area of orifice $A_o = \pi d_0^2 / 4$, and discharge correction factor $\varnothing$ is 0.8 if the lubricant is air. Considering that the flow is isothermal, the discharge coefficient $\varphi$ under both supersonic and subsonic conditions is calculated based on Equation (6) [38].

$$\varphi(\beta_i) = \begin{cases} \left[ \frac{k}{k-1} \left( \beta_i^{\frac{2}{k}} - \beta_i^{\frac{k+1}{k}} \right) \right]^{\frac{1}{2}}, & \beta_i > \beta_\alpha \\ \left[ \frac{k}{2} \left( \frac{2}{k+1} \right)^{\frac{k+1}{k-1}} \right]^{\frac{1}{2}}, & \beta_i \leq \beta_\alpha \end{cases} \tag{6}$$

where $k = 1.4$ if lubricant is air. $B_i = P_{di} / P_0$, where $P_{di}$ is the gas pressure at the $i$th orifice outlet. $B_\alpha = (2/k + 1)^{(k+1/k-1)}$. Conclusion could be obtained from Equations (5) and (6) that MFR of each section can be assessed by $\beta_i$. Based on the mass conservation law, MFR inside of 1-D gas film shown in Figure 2 can be written as

$$\dot{m}_i = b \int_0^{h_i} \rho u \, \mathrm{d}z \tag{7}$$

where $h_i = h_0 (1 - \varepsilon \cos \theta_i)$.

Equations (5) and (7) describe the mass conservation equations of the gas that flows through the bearing. Under isothermal conditions, the gas state equation is

$$\frac{P}{\rho} = \frac{P_a}{\rho_a} \tag{8}$$

Given that the velocity of the bearing's surface is relatively low, when $z = 0$ and $z = h_i$, $u$ is expected to be zero for simplicity. Subsequently, $u$ is calculated by double integration of Equation (4)

$$u = -\frac{1}{2\eta}\frac{dP}{dx}z(h_i - z) \tag{9}$$

Equation (9) is used to substitute $u$ in Equation (7). Then, MFR through a section with width $b$ is derived as

$$\dot{m}_i = \frac{\rho b h_i^3}{12}\frac{dP}{dx} \tag{10}$$

Considering Equation (8), $\rho$ can be put into Equation (10), and the variables $P$ and $x$ are separated,

$$PdP = \frac{12\eta\,\dot{m}_i P_a}{b h_i^3 \rho_a}x\ (0 < x < l) \tag{11}$$

Equation (10) is integrated; the bearing's pressure $P$ can be calculated as

$$P^2 - P_{di}^2 = -\frac{24\eta\,\dot{m}_i P_a}{b h_i^3 \rho_a}x \tag{12}$$

Given the boundary conditions, $P = P_a$ when $x = l$, Equation (12) can be written as

$$P_a^2 - P_{di}^2 = -\frac{24\eta\,\dot{m}_i P_a}{b h_i^3 \rho_a}l \tag{13}$$

Equation (12) is divided by Equation (13); $P$ is defined as

$$P = \sqrt{P_{di}^2 - (P_{di}^2 - P_a^2)\frac{x}{l}} \tag{14}$$

To obtain LCC, every section is considered separately. The pressure distribution in $i$th section is integrated,

$$F_i = \frac{D}{2}\int_{-\frac{\pi}{N}}^{\frac{\pi}{N}}\left[2\int_0^l Pdx + P_{di}(L - 2l)\right]\cos\theta\,d\theta \tag{15}$$

In every section, the composition of forces generated from pressure is symmetrical about the axial line on the gas film through $i$th orifice, which leads to

$$F_i = D\sin\frac{\pi}{N}\left[2\int_0^l Pdx + P_{di}(L - 2l)\right] \tag{16}$$

Equation (14) is substituted into Equation (16) and the integral value calculated,

$$F_i = DLP_0 K_i\sin\frac{\pi}{N},\ K_i = \beta_i\left[\frac{(L - 2l)}{L} + \frac{4}{3}\frac{l}{L}\frac{1 - (\frac{\sigma}{\beta_i})^3}{1 - (\frac{\sigma}{\beta_i})^2}\right] \tag{17}$$

where $\sigma = P_a/P_0$.

The direction of $F_i$ is at a certain angle $\theta_i = 2(i-1)\pi/N$ with the vertical direction, all those forces in the vertical direction are combined, LCC of the gas film can be expressed as

$$W = \sum_{i=1}^{N} F_i \cos\theta_i = DLP_0 \sin\frac{\pi}{N} \sum_{i=1}^{N} K_i \cos\theta_i \tag{18}$$

LCC coefficient (LCCE) can be expressed as

$$C_W = \frac{W}{DLP_0} = \sin\frac{\pi}{N} \sum_{i=1}^{N} K_i \cos\theta_i \tag{19}$$

It is LCC of unit area. When bearing clearance is equal everywhere $h_i = h_0$, $K_i$ in every section is equal and $C_w$ must be equal to zero, indicating that bearing bears nothing.

## 3. Calculation Procedures of the Simplified Calculation Method

$\beta_i$ is an unknown variable in Equation (17). However, the mass conservation law indicates the inflow MFR must equal to the outflow MFR, suggesting that the $\dot{m}_i$ in Equations (5) and (13) must be equal:

$$\frac{\beta_i^2 - \sigma^2}{\sigma\varphi} = \frac{\varnothing A_o N}{h_i^3}\frac{12\eta}{\pi}\sqrt{\frac{2}{P_a\rho_a}}\frac{2l}{D} \tag{20}$$

Supposing

$$\zeta_i = \frac{\beta_i^2 - \sigma^2}{\sigma\varphi} \tag{21}$$

and

$$f_{1i} = \frac{\varnothing A_o N}{h_i^3} \tag{22}$$

$$f_2 = \frac{12\eta}{\pi}\sqrt{\frac{2}{P_a\rho_a}} \tag{23}$$

$$f_3 = \frac{2l}{D} \tag{24}$$

The definitions of $f_{1i}$, $f_2$ and $f_3$ are very intuitive to understand: $f_{1i}$ is the gas channel coefficient related to film thickness and orifice diameter. $f_2$ is the lubricant physical coefficient only related to the physical properties of lubricant itself, which is a constant for specific kind of gas. $f_3$ is the bearing structural coefficient related to structural parameters. For every section, Equations (21)–(24), Equation (20) can be denoted as

$$\zeta_i = f_{1i}\, f_2\, f_3 \tag{25}$$

If atmospheric pressure, supply pressure, lubricant gas and structural parameters are stated, $\zeta_i$ and $\beta_i$ are determined by Equations (25) and (21), respectively. Subsequently, LCC and LCCE are calculated by incorporating $\beta_i$ in Equations (18) and (19).

Given that $\beta_i$ and $\zeta_i$ are associated with each other by a nonlinear univariate equation, lots of simile solution methods can be used to solve Equation (25). Figure 3 demonstrates the detailed procedures of simplified calculation method, where $K_w = \Delta W/\Delta h$ is the bearing stiffness along the radial direction.

Next, $W$, $Cw$ and $Kw$ are calculated to optimize the bearing's parameters $P_0$, $L$, $D$, $N$ and $A_0$. Figure 3 also demonstrates that simplified calculation method is easier to operate compared with CFD and FEM.

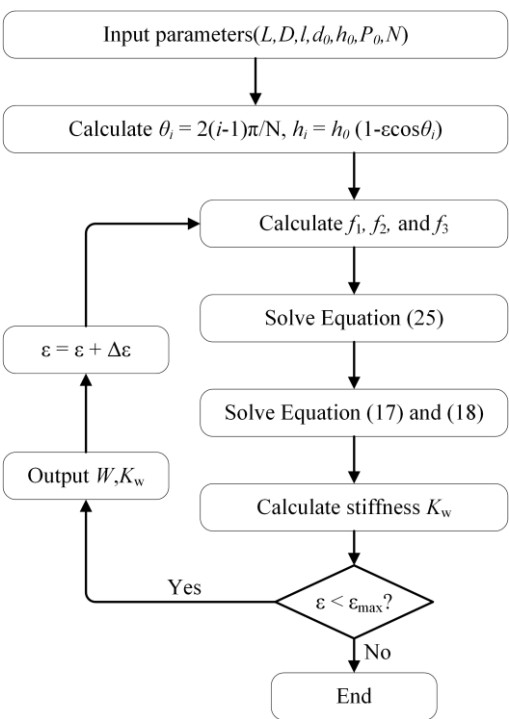

**Figure 3.** Calculation procedures of the simplified calculation method.

## 4. Application Case Study

To validate and evaluate this method, the calculation data of the simplified calculation method were compared to those of CFD obtained from [29]. This aerostatic spindle, named C200, has been widely used in actual production. The picture of a C200 and its section view are shown in Figure 4. The C200 spindle is a heavy-duty bearing, which is designed to carry a load of more than 1000 kg under the condition of double-end support. Thus far, we have produced more than 16 spindles, which have proven to be a successful product. Its earliest engineering design adopts the method described in this paper. Structural and gas-supply parameters are shown in Table 1. The LCC of the calculation examples of different eccentricities are shown in Table 2.

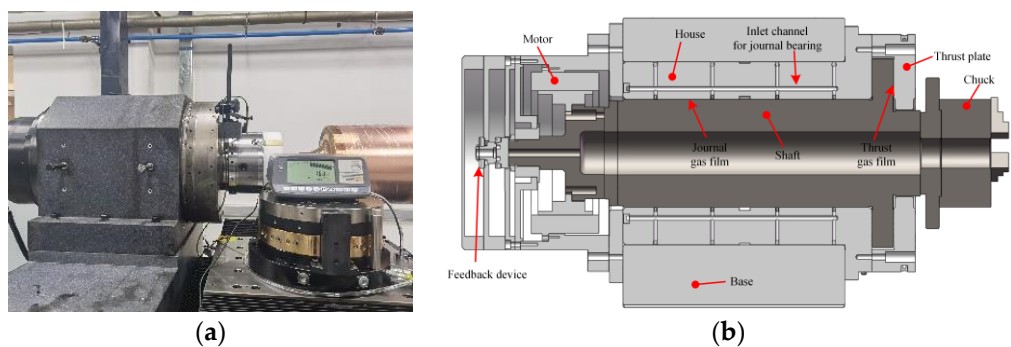

**(a)**　　　　　　　　　　　　　　　　　**(b)**

**Figure 4.** The aerostatic spindle and its cross-section view: (**a**) picture of the C200 spindle, (**b**) cross-section view and diagram of the C200 spindle.

**Table 1.** Parameters of the aerostatic journal bearing (C200).

| Term | Specification |
|---|---|
| Length, $L$ | 225 mm |
| Diameter of spindle rotor, $D$ | 200 mm |
| Orifice-end distance, $l$ | 65 mm |
| Orifice diameter, $d0$ | 0.2 mm |
| Gas film thickness, $h0$ | 20 μm |
| Supply pressure, $P_0$ | 0.5 MPa |
| Number of orifices in each row, $N$ | 12 |
| Lubricant | Clean air |

**Table 2.** LCC of the calculation example of different eccentricities (C200).

| Eccentricity Ratio | Eccentricity Value | CFD Result | Simplified Method Result | Ratio of Deviation |
|---|---|---|---|---|
| 0 | 0 μm | 0 | 0 | 0 |
| 0.1 | 2 μm | 1300 N | 1350 N | 3.8% |
| 0.2 | 4 μm | 2550 N | 2614 N | 2.5% |
| 0.3 | 6 μm | 3640 N | 3717 N | 2.1% |
| 0.4 | 8 μm | 4520 N | 4622 N | 2.3% |

Figure 5 shows the calculation results of CFD and the simplified method described in this paper under different eccentricities, which are 0, 0.25, and 0.5, respectively. The comparison shows that there are two main differences: (1) The geometric size of orifice is not considered in the one-dimensional flow model, so there is no raised high-pressure area in the cross-section; that is, the one-dimensional flow model cannot represent the diffusion flow after the orifice. (2) Since the diffusion flow is not considered, the flow resistance in the one-dimensional flow model is smaller than that in the total model of CFD. The orifice and gas film are somehow analogous to two nonlinear electric resistors, and the recess pressure is similar to the electric voltage between those two resistors. According to the Kirchhoff circuit law, the pressure after the orifice is smaller when calculated using the simplified one-dimensional model. Although the annular flow makes the calculated value of the one-dimensional flow model larger, the smaller pressure after the orifice makes the calculated value smaller; the superposition of those two effects makes the difference between the calculated values not very large.

Table 2 shows that the calculation results of the simplified calculation method are in good agreement with those of CFD. In addition, those computational results are graphically compared in Figure 6. LCC calculated by this method is slightly larger than CFD, and the relative error is within 5% in all the different eccentricities. Obviously, the assumption of neglecting circumferential flow shows little effect in this case because gas film thickness is about 1/10,000 of spindle diameter. This simplified calculation method can be applied because most of the aerostatic bearings have a film thickness of about 5~15 μm and are small enough compared with the bearing structural parameters.

In the calculation process, the most notable is the time taken by CFD and the simplified calculation method. In order to obtain the calculation results in Figure 6, CFD calculation took about 26 min and the simplified calculation method took only a few seconds; both programs were run on the same computer with an i7 CPU, not to mention the half-day of pre-modeling with CFD. Considering the calculation accuracy and efficiency, the method proposed in this paper is a very effective method in the early stage of engineering design.

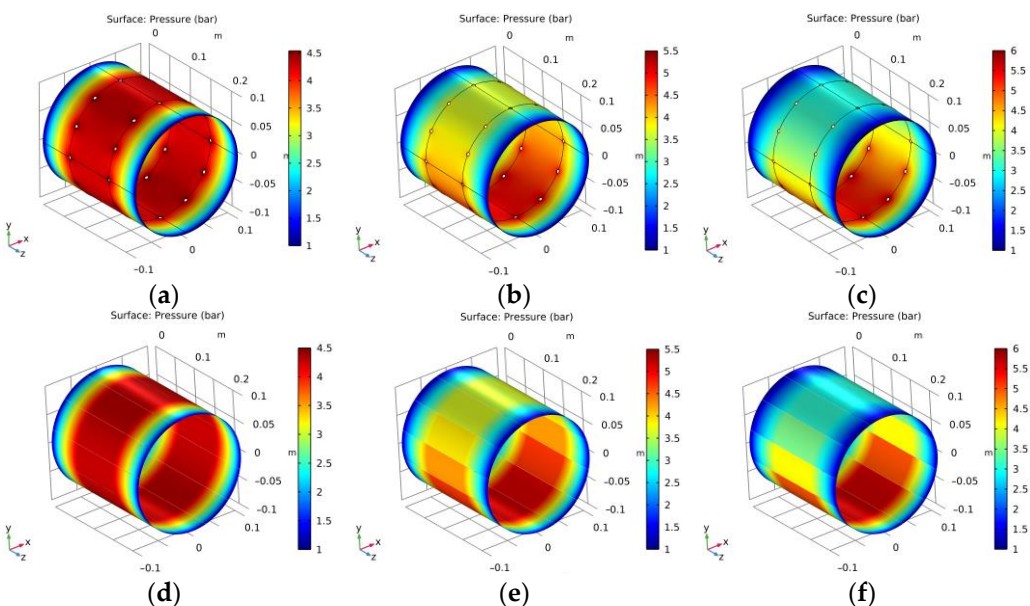

**Figure 5.** The aerostatic C200 spindle and air-pressure-distribution simulation results calculated using CFD and simplified method: (**a**) the CFD result without eccentricity, (**b**) the CFD result with eccentricity ratio of 0.25, (**c**) the CFD result with eccentricity ratio of 0.5, (**d**) the simplified method result without eccentricity, (**e**) the simplified method result with eccentricity ratio of 0.25, (**f**) the simplified method result with eccentricity ratio of 0.5.

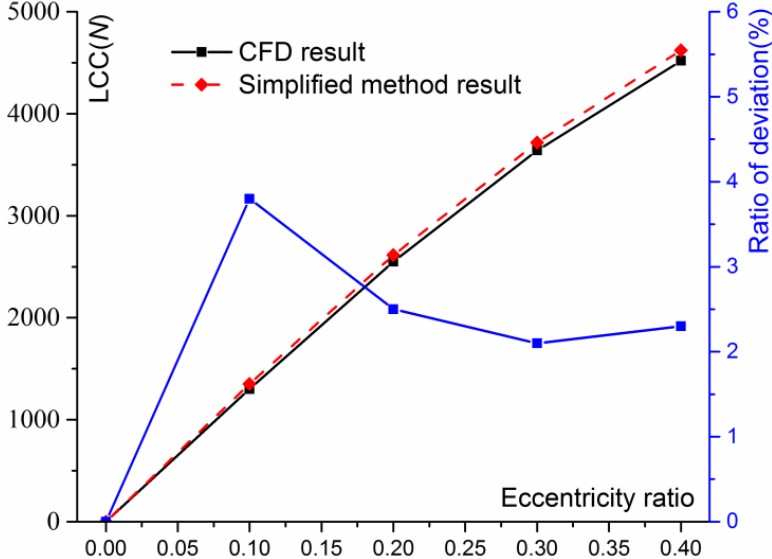

**Figure 6.** LCC calculated by CFD and simplified calculation method.

Figure 7 shows another widely used precision spindle in diamond-turning lathe, with a diameter of 100 mm, named C100, with their parameters listed in Table 3. Thus far, we have produced more than 50 spindles; in addition, its initial design was completed by the method proposed in this paper. The pressure-distribution cloud map calculated using CFD is similar to Figure 5c, shown in Figure 8.

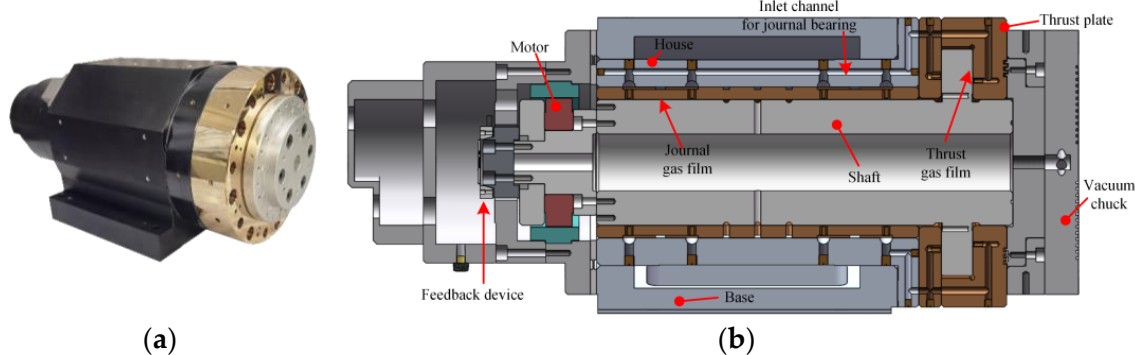

**Figure 7.** The aerostatic spindle used in diamond-turning lathes: (**a**) picture of spindle, (**b**) cross-section view and diagram of the C100 spindle.

**Table 3.** Parameters of the aerostatic journal bearing (C100).

| Term | Specification |
|---|---|
| Length, $L$ | 100 mm |
| Diameter of spindle rotor, $D$ | 100 mm |
| Orifice-end distance, $l$ | 25 mm |
| Orifice diameter, $d0$ | 0.2 mm |
| Gas film thickness, $h0$ | 15 μm |
| Supply pressure, $P_0$ | 0.5 MPa |
| Number of orifices in each row, $N$ | 8 |
| Lubricant | Clean air |

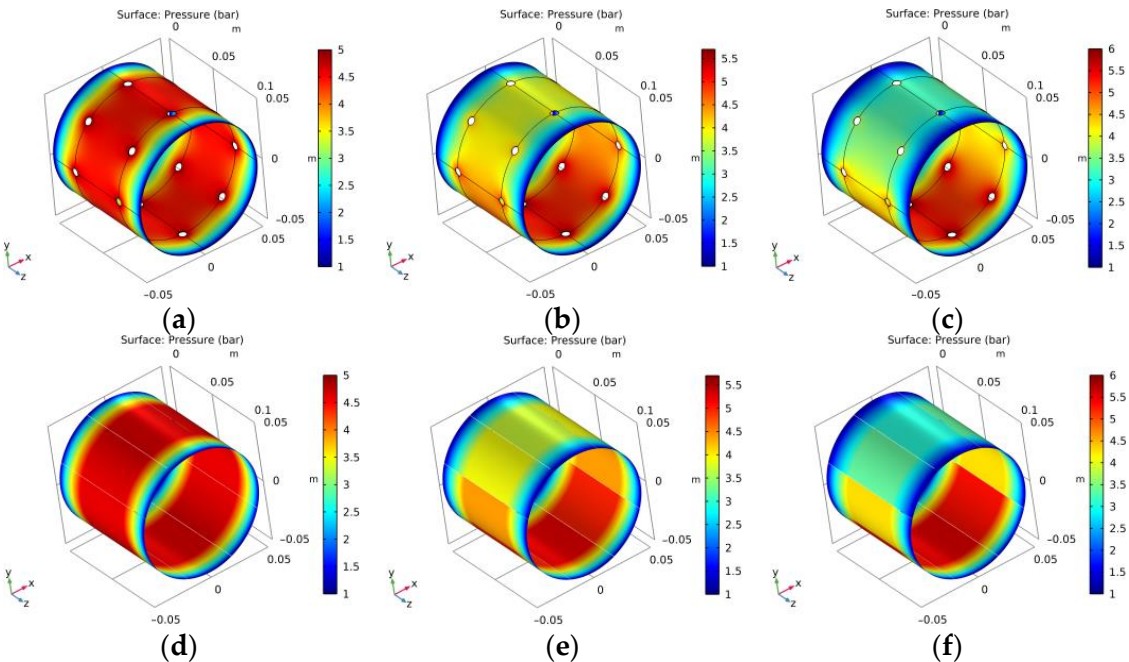

**Figure 8.** The aerostatic C100 spindle and air-pressure-distribution simulation results calculated using CFD and simplified method: (**a**) the CFD result without eccentricity, (**b**) the CFD result with eccentricity ratio of 0.25, (**c**) the CFD result with eccentricity ratio of 0.5, (**d**) the simplified method result without eccentricity, (**e**) the simplified method result with eccentricity ratio of 0.25, (**f**) the simplified method result with eccentricity ratio of 0.5.

It can be seen from Table 4 that for this spindle, the LCC difference between those two methods becomes larger, but still does not exceed 10%. The main reason for this difference is annular flow. When the radial bearing is eccentric, the gas will generate a flow from a high-pressure area with thin film to the low pressure area with thick film, resulting in a decrease in bearing capacity. The one-dimensional flow model introduced in this paper does not take into account the annular flow; therefore, it is expected that the simplified method will predict a higher LCC than the CFD method for the small diameter of the main shaft.

**Table 4.** LCC of the calculation example of different eccentricity (C100).

| Eccentricity Value | CFD Result | Simplified Method Result | Ratio of Deviation |
|---|---|---|---|
| 0 μm | 0 N | 0 N | 0 |
| 1.2 μm | 242.8 N | 266.2 N | 9.6% |
| 2.4 μm | 476.3 N | 514.3 N | 8.0% |
| 3.6 μm | 673.5 N | 730.6 N | 8.5% |
| 4.8 μm | 838.7 N | 912.2 N | 8.7% |
| 6.0 | 968.8 N | 1061 N | 9.5% |

The C100 and C200 have the same ratio between the orifice-end face and the entire bearing length, but the difference between the calculation results of C100 and C200 and the numerical analysis is not the same; the key lies in the orifices' number. The C100 has only eight restrictors on each circumference, the division of each area is rougher, and the diffusion effect and annular flow are more obvious, so the calculation results are quite different from CFD.

## 5. Impact of $\zeta_i$ on a Bearing's Performance

According to Equation (12), $P_d$ determines the pressure distribution in the gas film $P$ once the bearing's parameters have been provided, while $P_d$ is determined by $\zeta_i$ deduced from Equations (20)–(25). Furthermore, the effects of a bearing's parameters on its performance is affected by changing $\zeta_i$.

As demonstrated in Figure 9, the bearing's performance can be determined by analyzing the impact of $\zeta_i$ on $\beta_i$ according to Equation (21), $\beta_i = P_{di}/P_0$ and $\beta_i$ is determined by $\zeta_i$. At varying supply pressures (from $\sigma = 1/2$ to $\sigma = 1/10$), $\beta_i$ increases with the growth in $\zeta_i$ and tends to remain in the supersonic area when $\zeta_i$ is larger than 10. Figure 9 also highlights that it is relatively easier for $\zeta_i$ to have an inclination to be saturated in the subsonic area with the decrease in supply pressure. Gas should flow sub-sonically in order to avoid shock waves; otherwise, LCC will sharply decrease and there is hardly any stiffness. From the red line of Figure 9, where $\beta_i = 0.528$, it can be seen that the suitable range of the bearing's parameters is narrow at extremely high supply pressure ($\sigma < 1/8$). Therefore, 0.4~0.6 MPa is an appropriate supply pressure in most cases.

Generally, the ratio of $l$ to $L$ is equal to 1/4 or 1/5. When the ratio is 1/4, Figure 10 indicates the distribution of $K_i$ in Equation (17) with $\beta_i$ at different supply pressures. Obviously, $K_i$ is approximately linear to $\beta_i$ and almost independent of supply pressure. The distribution of $\zeta_i$ with $K_i$ is shown in Figure 11. Thus, it is speculated that $\zeta_i$ and $K_i$ have the same distribution to $\zeta_i$ and $\beta_i$. From the physical meaning of Equation (19), $K_i$ is proportional to the LCC of the $i$th section.

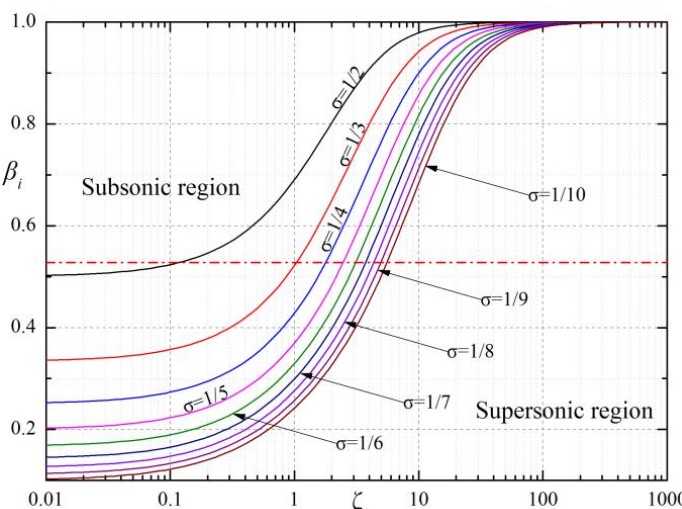

**Figure 9.** Effect of $\zeta_i$ on $\beta_i$.

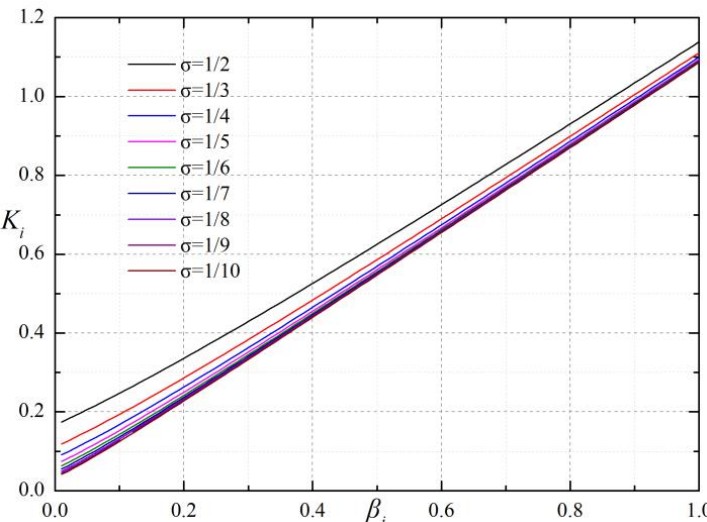

**Figure 10.** Effect of $\beta_i$ on $K_i$.

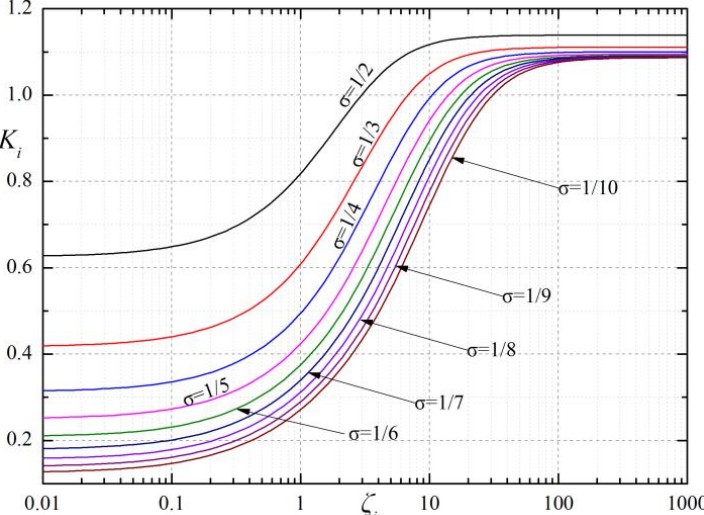

**Figure 11.** Effect of $\zeta_i$ on $K_i$.

Figure 12 demonstrates the influence of $\zeta_i$ on $K_{wi}$. Actually, the lines shown in Figure 12 are proportional to the derivative of lines shown in Figure 11. As Figure 12 indicates, the maximum stiffness exists when $\zeta_i$ is 3~10 at different varying supply pressures. With increasing supply pressures, $\zeta_i$ becomes larger after reaching the maximum stiffness. Compared with [8], this is the difference between a Cartesian coordinate system and cylindrical coordinate system.

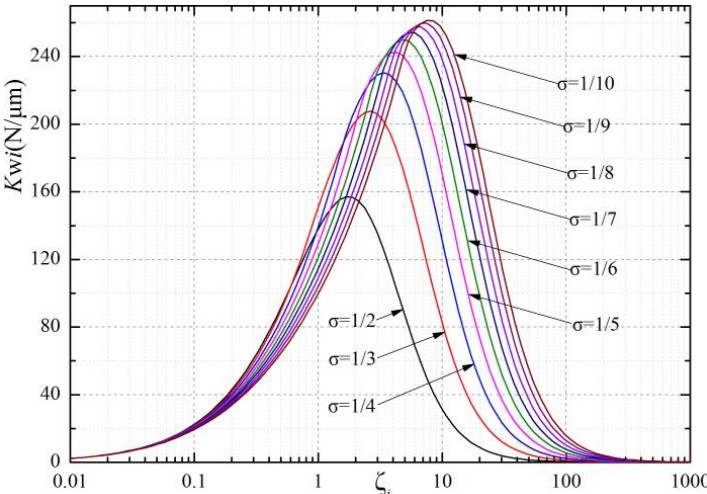

**Figure 12.** Effect of $\zeta_i$ on $K_{wi}$.

The line shapes displayed in Figure 13 are the same as those in Figure 12, except that the value is several times larger. In other words, a bearings' stiffness is directly connected with the stiffness in every section. It can be proved as follows:

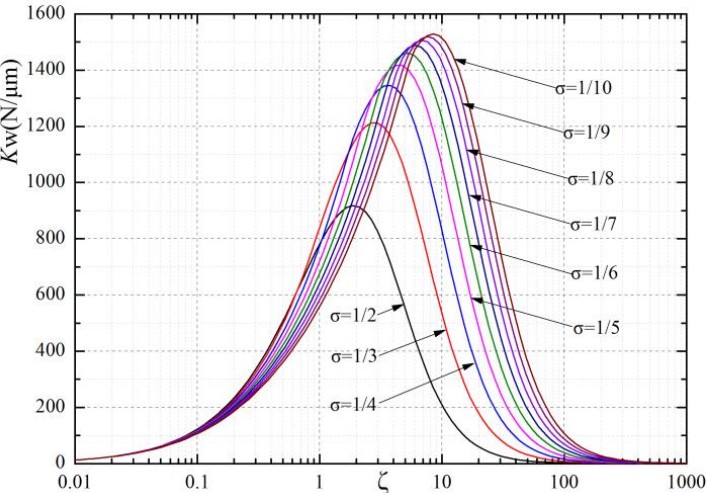

**Figure 13.** Effect of $\zeta_i$ on $K_w$.

The total stiffness $K_w$ is the vector sum of $K_{wi}$ in the vertical direction, that is

$$K_w = \sum_{i=1}^{N} K_{wi} \frac{\Delta h_i}{\Delta h} \cos \theta_i = \sum_{i=1}^{N} K_{wi} \cos^2 \theta_i \tag{26}$$

The partial derivation of both sides of Equation (26) is expressed as Equation (27).

$$\frac{\partial K_w}{\partial \Delta h} = \sum_{i=1}^{N} \frac{\partial K_{wi}}{\partial \Delta h} \cos^2 \theta_i \tag{27}$$

To have the left side equal to zero, every partial derivative ($i = 1 \sim N$) on the right side needs to be zero. To approach the maximal stiffness, every gas film section in the bearings must be designed for maximal stiffness.

The effect of $\zeta_i$ on MFR is shown in Figure 14. It is obvious that supply pressure has a significant influence on a bearings' MFR. A bearings' MFR will rise in pace with supply pressure. In addition, MFR tends to have a constant value when $\zeta_i$ is smaller than 1. With the increase in $\zeta_i$, MFR can decrease due to the flow-rate characteristics of an ideal orifice.

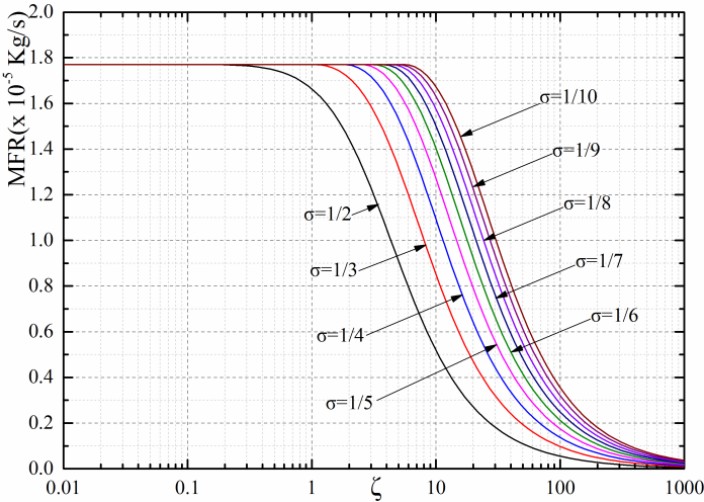

**Figure 14.** Effect of $\zeta$ on MFR.

## 6. Conclusions

In this study, a simplified calculation method of aerostatic journal bearing with MPOTRs was presented. The calculation procedure was summarized and can be programmed easily to analyze the performance of two-row-orifice type journal aerostatic bearings, which simplifies the analysis process of aerostatic journal bearing's performance prediction and offers a straightforward way to optimize parameters. In addition, this method was verified with computational fluid dynamics by two actual case studies, and it was found that the LCC difference between those two methods is less than 5% for a 200 mm-diameter spindle, and less than 10% for a 100 mm-diameter spindle.

In order to satisfy the critical need for high stiffness and load carrying capacity, every gas film section in bearings should be designed to reach its maximal stiffness. From the calculation process, it was found that $\zeta_i$ is a critical factor that influences bearings' performance through the equations derived from this method. The following conclusions can be reached by analyzing the association between $\zeta_i$ and performance parameters:

(1) $\zeta_i$ is a crucial parameter of the bearing's performance. It can be assessed by the discharge correction factor $\varnothing$, sectional area of the orifice $A_0$, orifice number $N$, gas film thickness in the $i$th section $h_i$, kinematic viscosity of air $\eta$, atmospheric density $\rho_a$, atmospheric pressure $P_a$, journal bearing diameter $D$, distance between orifices and gas film edge $l$, and journal bearing length $L$.

(2) $\zeta_i$ is the product of $f_1$ (a gas-channel coefficient), $f_2$ (a gas-lubricant coefficient) and $f_3$ (a bearing structural coefficient). A bearing with better performance, such as LCC, stiffness and MFR, could be obtained if $\zeta_i$ is in the range of $1 \sim 10$.

(3) With increasing $\zeta_i$, a bearing's stiffness in each section will reach a maximal value and then decrease. Here, 0.4~0.6 MPa is an appropriate supply pressure range in most cases, as the bearing's parameters are narrow at extremely high supply pressures to avoid shock waves.

**Author Contributions:** Conceptualization, Y.W., J.X. and B.W.; Methodology, Y.W. and Z.Q.; Software, Y.W. and J.X.; Validation, Y.W.; Formal Analysis, W.C.; Investigation, Y.W. and J.X.; Resources, Y.W. and B.W.; Data curation, Y.W. and J.X.; Writing—original draft preparation, J.X.; Writing—review and editing, Y.W. and W.C.; Visualization, Y.W. and J.X.; Supervision, Z.Q. and B.W.; Project administration, Z.Q. and B.W.; Funding acquisition, Z.Q. and B.W. All authors have read and agreed to the published version of the manuscript.

**Funding:** The work was supported by the Open Project Program of State Key Laboratory of applied optics (SKLAO2021001A05); National Natural Science Foundation of China (No.51905130); and Heilongjiang Provincial Natural Science Foundation of China (No. LH2020E039).

**Data Availability Statement:** The data supporting reported results by the authors can be sent by e-mail.

**Conflicts of Interest:** The authors declare no conflict of interest.

## Nomenclature

| | |
|---|---|
| $\beta_i$ | Pressure ratio in $i$th section, $P_{di}/P_0$ |
| $\beta_\alpha$ | Critical pressure ratio |
| $\eta$ | Kinematic viscosity of air |
| $\rho$ | Density of gas in the bearing |
| $\rho_a$ | Atmospheric density |
| $\sigma$ | Ratio of pressure, $P_a/P_0$ |
| $\varepsilon$ | Eccentricity ratio |
| $\varnothing$ | Discharge correction factor |
| $\varphi$ | Discharge coefficient |
| $\zeta_i$ | Product of three coefficients |
| $A_0$ | Sectional area of the orifice |
| $b$ | Width of 1-D gas film |
| CFD | Computational fluid dynamics |
| $d_0$ | Diameter of the orifice |
| $D$ | Journal bearing diameter |
| $f_{1i}$ | Gas channel coefficient |
| $f_2$ | Lubricant physical coefficient |
| $f_3$ | Bearing structural coefficient |
| FEM | Finite element method |
| $h_0$ | Designed gas film thickness |
| $h_i$ | Gas film thickness in ith section |
| $\Delta h$ | Film thickness changing value |
| $k$ | Gas specific heat ratio |
| $K_w$ | Bearing stiffness |
| $l$ | Distance between orifices and gas film edge |
| $L$ | Journal bearing length |
| LCC, W | Load carrying capacity |
| LCCE, Cw | LCC coefficient |
| MFR | Mass flow rate |
| MPOTRs | Multiple pocketed orifice-type restrictors |
| $\dot{m}$ | Mass flow rate through the orifice |
| $\dot{M}$ | Bearing mass flow rate |
| $N$ | Orifice number |
| $P$ | Pressure in the bearing |
| $P_a$ | Atmospheric pressure |
| $P_d$ | Pressure at orifice outlet |
| $P_0$ | Supply pressure |

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
