# Peer review of "Evaluation and Application of an Engineering Calculation Method of the Static Performance of an Aerostatic Journal Bearing with Multiple Orifice-Type Restrictors"

_lubricants, doi:10.3390/lubricants10120332_

Round 1

Reviewer 1 Report

The author tried to develop a new method on the static performance of 2 aerostatic journal bearing with multiple orifice-type restrictors. The paper has some important messages to communicate. But it is not acceptable at its current state. A major revision is required and sufficient justification should be given to accept how such a method is superior to the existing FEM/ CFD simulations. The following concerns are raised in the review process and must be addressed in the revised manuscript.

1.      Line 31 please see if this format is allowed as per Lubricant guidelines, Lo and colleagues [4]. Also, check for other places

2.      Please do not use the SCM, simplified calculation method in the text. Give the full words.

3.      The highlights of the result should be there in the abstract. Please re-write the abstract.

4.      ‘This method is much easier and more intuitive compared with numerical 13 computational methods such as finite element method (FEM) and CFD’ such qualitative statements should not be there in the abstract.

5.      In the flow chart given in figure 3, eq.25 is mentioned first and then Eq. (17), (18). Will it not be better to rearrange the Equations such the earlier one will come first in the computation?

6.      The author should make sure that the cavitation effect and heat transfer through the fluid-structure interface are addressed in the model. As I am not able to see the thermal parameters or heat transfer coefficient in the equations, which plays a measure role in the lubrication process. The authors should clarify.

7.      How the boundary conditions are dealt with  the new method.

8.      More discussions on how  such model is advantageous should be elaborated.

9.      Please improve the conclusion.

10.   Please improve the English of the whole manuscript, especially discussions should be revisited for more clarity sentences.

11.   Number of references offered is the bare minimum in the case of a journal paper. Please add more research references (At least 25 for an acceptable manuscript.) Accordingly please revise the Introduction.

12.   Results obtained in CFD and in the SCM must be graphically compared and this method is thus justified as an acceptable alternative.

Author Response

 The point-by-point response to the reviewer’s comments has been attached to a word file. Please refer to this file, thanks.

Reviewer 2 Report

The authors have conducted some studies for the performance analysis of aerostatic journal bearing with multiple orifice-type restrictors. The major problems that are listed as following must be solved by authors.

In the manuscript, the recent studies and applications on methods for calculating the performance of aerostatic bearings were missed out.

The authors claimed that a simplified calculation method was proposed in this manuscript. However, the method is a commonly used engineering method that has been fully discussed in the published literatures.

The authors claimed to have validated this simplified method by CFD. However, the reviewer does not find the CFD simulation results in this paper. There was only a data table. This is very confusing.

Author Response

Q: The authors have conducted some studies for the performance analysis of aerostatic journal bearing with multiple orifice-type restrictors. The major problems that are listed as following must be solved by authors. In the manuscript, the recent studies and applications on methods for calculating the performance of aerostatic bearings were missed out.

A: The introduction has been greatly updated. Some recent development of calculating the performance of aerostatic bearings has been added and summarized.

Q: The authors claimed that a simplified calculation method was proposed in this manuscript. However, the method is a commonly used engineering method that has been fully discussed in the published literatures.

A: The commonly used method in practical engineering is the gauge pressure ratio method proposed by Powell et al, which is detailed described in book: Powell, John William. Design of aerostatic bearings. Machinery publishing, 1970. The gauge pressure ratio method borrows the symbol of hydrostatic bearings, and the compressibility of gas is not properly handled, so it is necessary to manually look up the table to obtain the bearing characteristics.

The true idea comes from the literature: Li, Yuntang, and Han Ding. "A simplified calculation method on the performance analysis of aerostatic thrust bearing with multiple pocketed orifice-type restrictors." Tribology international 56 (2012): 66-71. Li et.al proposes a similar calculation method but only deals with the thrust bearing. There is of great difference between the boundary conditions between thrust bearing and journal bearing, therefore, the 1-D flow model is quite different for thrust and journal bearings. The simplified model presented in this paper is a good way to deal with engineering aerostatic radial bearings.

Q: The authors claimed to have validated this simplified method by CFD. However, the reviewer does not find the CFD simulation results in this paper. There was only a data table. This is very confusing.

A: A major advantage of the engineering simplification method over the CFD algorithm is the ability to calculate bearing characteristics for a large number of design parameters in a short time. When changing design parameters, CFD needs to re-mesh, set boundary conditions, form and solve matrices, and display post-processing results, also it is accompanied by a lot of manual labor. Therefore, in this paper, two examples of radial bearings in engineering design practice are calculated and the differences between the proposed method and the CFD calculation results are compared. The calculation results show that the calculation accuracy of this simplified method is sufficient for preliminary engineering designs.

Of course, the authors admit that it is not intuitive to present the results in just one table. Therefore, the relevant results have been presented in a coordinate graph in the revised version.

Reviewer 3 Report

The paper presents some interesting and innovative work on developing a simplified calculation method on the static performance of the aerostatic journal bearing with multiple orifice-type restrictors. The work is further supported by design modelling and analysis, and some computational results. However, the paper manuscript needs to undertake the following minor revisions:

(1) ζi should be listed in 'Nomenclature'.

(2) The paper should better include a section of 'Application case study', to further illustrate how the simplified calculation method (SCM) being applied in 'real' design applications (of aerostatic bearing spindles and/or slideways (or rotary tables).

(3) In Section 3, the paper should provide further clarification and discussion on the configuration and design details of the orifices, and how the proposed SCM taking account of these factors (i.e., the configuration and design details of the orifices).

(4) The following very relevant paper in the topic area should be better included in References section, particularly against above comments (3) and (2):

CFD based investigation on influence of orifice chamber shapes for the design of aerostatic thrust bearings at ultra-high speed spindles, Tribology International, Vol. 92, 2015, pp. 211–221.

Author Response

The paper presents some interesting and innovative work on developing a simplified calculation method on the static performance of the aerostatic journal bearing with multiple orifice-type restrictors. The work is further supported by design modelling and analysis, and some computational results. However, the paper manuscript needs to undertake the following minor revisions:

(1) ζi should be listed in 'Nomenclature'.

A: ζi has been added in 'Nomenclature'.

(2) The paper should better include a section of 'Application case study', to further illustrate how the simplified calculation method (SCM) being applied in 'real' design applications (of aerostatic bearing spindles and/or slideways (or rotary tables).

A: The ‘Application case study’ section has been added. In this section, two typical journal bearings adopting two-row orifices are described and analyzed by the simplified calculation method and CFD, respectively. Also, the calculation time was compared, which proves this simplified calculation method offers sufficient accuracy in practical engineering but cost less time.

(3) In Section 3, the paper should provide further clarification and discussion on the configuration and design details of the orifices, and how the proposed SCM taking account of these factors (i.e., the configuration and design details of the orifices).

A: The typical location of two-row-orifice aerostatic journal bearings has been plot in figure 1. Also, it can be more intuitive to see how those orifices are distributed in the pressure distribution diagram in Figure 4.

(4) The following very relevant paper in the topic area should be better included in References section, particularly against above comments (3) and (2):

- CFD based investigation on influence of orifice chamber shapes for the design of aerostatic thrust bearings at ultra-high speed spindles, Tribology International, Vol. 92, 2015, pp. 211–221.

A: This paper is cited to guide reader to find a detailed description of orifice.

Round 2

Reviewer 1 Report

The authors made significant corrections in the manuscript and now it is suitable for publication.  

Author Response

Thanks for your kind comments. English language and style are minor spell checked.

Reviewer 2 Report

The author has revised the manuscript, but the following problems exist:

(1)    The authors added references to improve the literature review. However, the reviewer did not find articles published in the last two years (2021 and 2022).

(2)    The authors stated that the method is “a fast calculation method to design radial bearings in engineering”, which is an existing method according to the opinion of the reviewer. So the authors should indicate the earliest documented source of the method and cite the literature.

(3)    The manuscript is titled as A simplified calculation method on the static performance of aerostatic journal bearing with multiple orifice-type restrictors, but the method is not original to the author. The author's description of the method in the paper is not different from that in the existing literature. In other words, the reviewer does not find the authors' contribution to the method in the paper, and the use of this title is not appropriate.

(4)    The authors provided pressure distributions obtained by CFD methods to validate the simplified method. In this case, the author should provide the pressure distribution diagram calculated by the simplified method. However, the reviewer does not find any diagram of pressure distributions calculated by the simplified method.

(5)    What the authors analyze in Section 4 is very similar to the published literature on gas lubrication research. The authors should explain the differences between what is analyzed in that section and the existing available literature.

Author Response

The author has revised the manuscript, but the following problems exist:

(1)    The authors added references to improve the literature review. However, the reviewer did not find articles published in the last two years (2021 and 2022).

A: The introduction has been updated. Several recent literature in the last two years of calculating the performance of aerostatic bearings is added and summarized.

(2)    The authors stated that the method is “a fast calculation method to design radial bearings in engineering”, which is an existing method according to the opinion of the reviewer. So the authors should indicate the earliest documented source of the method and cite the literature.

A: As stated in the introduction section, the commonly used method in practical engineering is the gauge pressure ratio method proposed by Powell et al, which is detailed described in book: Powell, John William. Design of aerostatic bearings. Machinery publishing, 1970. The gauge pressure ratio method borrows the symbol of hydrostatic bearings, and the compressibility of gas is not properly handled, so it is necessary to manually look up the table to obtain the bearing characteristics.

The method similar to this article is proposed in the literature: Li, Yuntang, and Han Ding. "A simplified calculation method on the performance analysis of aerostatic thrust bearing with multiple pocketed orifice-type restrictors." Tribology international 56 (2012): 66-71. In which Li et.al proposes a one-dimensional simplified model for multi-orifice thrust bearings. Since there is of great difference between the boundary conditions between thrust bearing and journal bearing, therefore, the 1-D flow model is quite different for thrust and journal bearings. The simplified model presented in this paper is a good way to deal with engineering aerostatic radial bearings.

A review of the literature shows that the method mentioned in this paper has been proposed in the early stage of the development of lubrication mechanics, but it is limited by computer performance and lacks further analysis of key processes. According to the reviewer's request, the earliest 2 papers were cited:

  1. Tang, I.C.; Gross, W.A. Analysis and Design of Externally Pressurized Gas Bearings. Tribol Trans 1962, 5, 261-284.
  2. Jason R. L. Analytical and Experimental Study of Externally Pressurized Air Lubricated Journal Bearings. J Basic Eng 1962, 84, 159-165.

(3)    The manuscript is titled as “A simplified calculation method on the static performance of aerostatic journal bearing with multiple orifice-type restrictors”, but the method is not original to the author. The author's description of the method in the paper is not different from that in the existing literature. In other words, the reviewer does not find the authors' contribution to the method in the paper, and the use of this title is not appropriate.

A: The title has been changed to ‘Evaluation and application of an engineering calculation method on the static performance of aerostatic journal bearing with multiple orifice-type restrictors’. Also, there are two main highlights in this paper: One is to compare the difference between this simple method and numerical simulation for the first time, and give a simple explanation. One the other hand, this paper discusses the influence of a key parameter on the static performance of journal bearings’ parameters.

 (4)    The authors provided pressure distributions obtained by CFD methods to validate the simplified method. In this case, the author should provide the pressure distribution diagram calculated by the simplified method. However, the reviewer does not find any diagram of pressure distributions calculated by the simplified method.

A: The pressure distributions calculated by the simplified method for C200 and C100 spindle under different conditions have been provided in the section of ‘Application case study’.

(5)    What the authors analyze in Section 4 is very similar to the published literature on gas lubrication research. The authors should explain the differences between what is analyzed in that section and the existing available literature.

A: The main function of Section 4 is to compare the performance difference between the CFD and the method described in the paper, with two kinds of spindle developed in our laboratory, aiming to verify the effectiveness of this simplified method (On the basis of verifying the effectiveness of this method, the influence of its design parameters on the final performance can be analyzed in Section 5.). Therefore, some conclusions are reflected in other literatures since basic rules of aerostatic bearings are the same. Other than that, this section gives an explanation why the one-dimensional flow model can give relatively accurate results. Please see Section 4 for more information.

Round 3

Reviewer 2 Report

The authors adopted the suggestions of reviewer and revised the manuscript. The revised manuscript is acceptable.